# Digital delivery of behavioural activation therapy to overcome depression and facilitate social and economic transitions of adolescents in South Africa (the DoBAt study): protocol for a pilot randomised controlled trial

Bianca D Moffett ![ORCID],[1] Julia R Pozuelo ![ORCID],[1,2] Alastair van Heerden ![ORCID],[3,4] Heather A O'Mahen,[5] Michelle Craske,[6,7] Tholene Sodi ![ORCID],[8] Crick Lund ![ORCID],[9,10] Kate Orkin ![ORCID],[11] Emma J Kilford ![ORCID],[12] Sarah-Jayne Blakemore ![ORCID],[1,13] Mahreen Mahmud,[14] Eustasius Musenge,[1,15] Meghan Davis,[2] Zamakhanya Makhanya,[1] Tlangelani Baloyi,[16] Daniel Mahlangu,[1] Gabriele Chierchia ![ORCID],[13] Sophie L Fielmann,[13] F Xavier Gómez-Olivé ![ORCID],[1] Imraan Valodia ![ORCID],[17] Stephen Tollman ![ORCID],[1,18] Kathleen Kahn ![ORCID],[1,18] Alan Stein ![ORCID] [1,2]

BDM and JRP are joint first authors.
KK and AS are joint senior authors.

For numbered affiliations see end of article.

**Correspondence to**
Dr Bianca D Moffett;
bianca.moffett@wits.ac.za

## ABSTRACT

**Introduction** Scalable psychological treatments to address depression among adolescents are urgently needed. This is particularly relevant to low-income and middle-income countries where 90% of the world's adolescents live. While digital delivery of behavioural activation (BA) presents a promising solution, its feasibility, acceptability and effectiveness among adolescents in an African context remain to be shown.

**Methods and analysis** This study is a two-arm single-blind individual-level randomised controlled pilot trial to assess the feasibility, acceptability and initial efficacy of digitally delivered BA therapy among adolescents with depression. The intervention has been coproduced with adolescents at the study site. The study is based in the rural northeast of South Africa in the Bushbuckridge subdistrict of Mpumalanga province. A total of 200 adolescents with symptoms of mild to moderately severe depression on the Patient Health Questionnaire Adolescent Version will be recruited (1:1 allocation ratio). The treatment group will receive BA therapy via a smartphone application (the Kuamsha app) supported by trained peer mentors. The control group will receive an enhanced standard of care. The feasibility and acceptability of the intervention will be evaluated using a mixed methods design, and signals of the initial efficacy of the intervention in reducing symptoms of depression will be determined on an intention-to-treat basis. Secondary objectives are to pilot a range of cognitive, mental health, risky behaviour and socioeconomic measures; and to collect descriptive data on the feasibility of trial procedures to inform the development of a further larger trial.

## STRENGTHS AND LIMITATIONS OF THIS STUDY

⇒ The intervention has been iteratively coproduced with local adolescents using multiple user-centred design methods to ensure that it was engaging, culturally relevant and usable for the targeted population.
⇒ We will use a rigorous mixed methods design to assess the feasibility, acceptability and initial efficacy of the digital intervention.
⇒ Culturally adapted measures of social–affective cognition, as well as a range of relevant mental health, risky behaviours and socioeconomic measures, will be piloted.
⇒ We have excluded participants with severe depression and high-risk suicidal ideation.
⇒ This study uses the Patient Health Questionnaire Adolescent Version, a validated screening tool to screen for depression, but does not provide a clinical diagnosis of depression.

**Ethics and dissemination** This study has been approved by the University of the Witwatersrand Human Research Ethics Committee (MED20-05-011) and the Oxford Tropical Research Ethics Committee (OxTREC 34-20). Study findings will be published in scientific open access peer-reviewed journals, presented at scientific conferences and communicated to participants, their caregivers, public sector officials and other relevant stakeholders.
**Trial registration numbers** This trial was registered on 19 November 2020 with the South African National Clinical



Trials Registry (DOH-27-112020-5741) and the Pan African Clinical Trials Registry (PACTR202206574814636).

## INTRODUCTION

Despite growing acknowledgement of the importance of adolescent mental health and the potential of investing in this formative period, it remains severely neglected.[1] This is particularly true in low-income and middle-income countries (LMICs), where 90% of the world's adolescents live, socioeconomic adversities affecting mental health are prevalent and mental health resources remain poor.[2] Given that most mental disorders have their onset before 25 years of age, early identification and treatment of emergent mental disorders has the potential to reduce chronicity and sequelae for individuals, and be an efficient strategy for addressing population-level mental health.[3] Investments in mental health have the potential to support socioeconomic transitions and contribute to breaking the cycle of poverty and mental ill-health in LMICs.[4]

Adolescence is a critical developmental period during which individuals develop their self-identity, acquire skills and preferences with which they navigate future challenges, and make decisions that can affect their long-term health, education, relationships and employment prospects.[5][6] Higher-order cognitive functions such as executive function and social cognition, which regulate the ability to strategise, set and maintain goals, and build successful relationships with others, develop significantly during this period.[7] Thus, the physical, cognitive, social and emotional capabilities acquired during adolescence lay a foundation for the well-being of individuals throughout their adult lives.[8] Depression interferes with the acquisition of these capacities, thus limiting young people's ability to fulfil their potential.

Globally, depression is one of the leading causes of disability among adolescents, and suicide is the third leading cause of death among the 15–19 year age group.[2][9] Left untreated, depression affects interpersonal relationships, interferes with schooling and disrupts productivity.[10] It also has a marked negative effect on executive function and social cognition, with depressed individuals consistently underperforming on cognitive assessments compared with healthy controls.[11][12] Furthermore, depression in adolescence has been associated with a greater risk for substance use, poor sexual health, delinquency and a significant reduction in future income.[13][14] Scalable psychological therapies to address depression among adolescents, particularly in low-resource contexts, are urgently needed.

Several barriers hinder progress in providing treatment to adolescents with depression in LMICs. First, while various treatments for depression have been shown to be effective among adolescents in high-income contexts, few have been adapted for or rigorously evaluated in low-resource or culturally diverse settings.[15][16] Second, there is a major shortage of mental health professionals in most LMICs,[17] and although task-sharing approaches present a promising strategy, they require substantial inputs in training and supervision of non-specialist healthcare workers.[18] Thirdly, there are significant challenges associated with relying on an already overburdened healthcare sector, not least that depression is rarely detected in primary care in most LMICs.[19] Finally, even in contexts where adolescent mental health services are available, stigma, lack of mental health awareness and issues with the acceptability of services prevent adolescents from seeking care.[20]

Digitally delivered behavioural activation (BA) therapy offers the potential to overcome a number of these barriers. BA is a highly effective psychological therapy that is easier to deliver and less costly than cognitive–behavioural therapy.[21][22] Importantly, there is also evidence that it can be effectively adapted for use: (1) with adolescents,[23][24] (2) in low-resource contexts and diverse cultural settings[25–27] and (3) in a digital format.[28] Delivering BA in a digital format may be particularly helpful in contexts where there is a paucity of mental health specialists and health systems are already juggling multiple competing priorities. It also offers the potential to overcome internalised barriers to care, such as stigma and lack of mental health awareness, and this form of guided self-help may be more acceptable to adolescents.[20]

Despite the many potential advantages of digital mental health interventions, studies that have evaluated their effectiveness among adolescents have yielded mixed results.[29][30] Most of the evidence on the effectiveness of digitally delivered BA comes from studies conducted in high-income countries,[28] and it is, therefore, an open question whether it is possible to generalise these findings to an African context, where conditions and resources differ vastly.[29][31] Furthermore, studies of online self-help interventions among adults in LMICs showed high attrition rates,[31] and adherence to mental health apps in all contexts is often low.[32] While some commercial smartphone apps attract more users, many have not been rigorously evaluated and show little fidelity to evidence-based treatments.[33][34]

This study will address an important gap in the literature by providing evidence on the feasibility, acceptability and initial efficacy of using digitally delivered BA to address depression among adolescents in an LMIC context. Furthermore, we will pilot a range of relevant mental health, risky behaviour and socioeconomic measures, as well as novel culturally adapted measures of executive function and social cognition.

### Study objectives

The coprimary objectives of this pilot study are: (1) to determine the feasibility and acceptability of a digitally delivered BA therapy intervention for adolescents living with depression in the Bushbuckridge subdistrict of Mpumalanga province, South Africa and (2) to provide preliminary evidence on any signals of initial efficacy (direction and magnitude) of any effects of the

intervention on depressive symptoms among adolescents in the intervention arm compared with control.

The secondary objectives are: (1) to pilot a range of mental health, social–affective cognition, risky behaviours and socioeconomic measures and (2) to collect descriptive data on trial procedures such as recruitment, retention, data collection, randomisation and blinding to inform key parameters in the development of a further large-scale trial.

## METHODS AND ANALYSIS

This protocol is reported in accordance with the Standard Protocol Items: Recommendations for Intervention Trials 2013 statement.[35] The statistical aspects of the pilot study are summarised here with details fully described in a statistical analysis plan (SAP) that is available in the online supplemental material.

### Study design

The DoBAt Study is a two-arm single-blind individual-level randomised controlled pilot feasibility trial. A total of 200 adolescents will be recruited (1:1 allocation ratio).

### Study setting

The study will be based in the Bushbuckridge subdistrict of Mpumalanga province, South Africa. While South Africa is classified as a middle-income country, large inequalities exist, and the study setting represents a rural area characterised by socioeconomic disadvantage and high rates of youth unemployment. Since 1992, the MRC/Wits Rural Public Health and Health Transitions Research Unit (MRC/Wits-Agincourt) has collected population data, with vital events (pregnancy outcome, deaths, in-migration and out-migration) and household composition updated annually through its Health and socio-Demographic Surveillance System. The total population under surveillance is currently ~116 000 inhabitants residing in 31 contiguous villages.[36] Recruitment will take place at schools within the study site. Notably, a previous study among female adolescents in this setting reported a prevalence of depressive symptoms of 18.2%[37] and yet access to treatment is severely limited. Similar data for male adolescents is not available.

### Patient and public involvement

Adolescents and members of the public have been involved at several stages of the study. The intervention was developed through extensive formative research and user-centred design with adolescents in the study area. We also conducted participatory workshops with adolescents and met with local educators, healthcare workers and relevant non-governmental organisations to obtain their input on trial methods, including recruitment and risk management strategies. We will seek the involvement of adolescents and members of the public in developing appropriate methods to disseminate study findings.

### Eligibility criteria

To be eligible for inclusion in the pilot trial participants must: (1) be between 15 and 19 years of age and in grades 9–11 at the beginning of the study; (2) have symptoms of mild to moderately severe depression indicated by a score between 5 and 19 on the Patient Health Questionnaire Adolescent Version (PHQ-A); (3) be able to read sufficiently in the local language (Xitsonga) to use the Kuamsha app; (4) intend to continue living in the study site for 12 weeks after the baseline assessment; and (5) provide written informed assent/consent to participate in the study, as well as parent/guardian consent if younger than 18 years.

Participants will be excluded if they: (1) have symptoms of severe depression as indicated by a score of >19 on the PHQ-A; (2) have current suicidal ideation with specific plans and means identified; (3) are receiving psychological treatment for a mental health condition at the time of enrolment; (4) have been hospitalised for at least 5 days for a severe psychiatric illness (specifically bipolar mood disorder, schizophrenia or other psychotic disorders) or life-threatening or other serious medical illness; (5) have a history of bipolar mood disorder, schizophrenia or other psychotic disorders; (6) lack capacity to consent to treatment or research participation or to use the app. Participants excluded for points 1 and 2 will be assessed by the risk management team and referred to local clinical services as per prior arrangements with local service providers.

### Intervention

All participants in the intervention and control arms will be given the entry-level Samsung Galaxy A2 Core Android smartphone, which they can keep at the end of the study. Furthermore, participants in both groups will receive 200 MB of mobile internet data at six different time points (0, 2.5, 5, 7.5, 11 and 24 weeks) to ensure they have data to use the app and complete the online surveys. Participants in both arms will receive active symptom monitoring via text messages sent to the smartphone every 2.5 weeks. Any adolescents who develop severe depression or high-risk suicidal ideation will be assessed by the risk management team (further details provided in online supplemental table 1) and referred to local clinical services. Adolescents started on antidepressant medication or who receive psychological therapy because of these referrals will not be discontinued, but we will take note of concomitant care and will examine this using sensitivity analyses.

### Control arm: enhanced standard of care

The non-intervention arm will receive a control app (the Kuchunguza app) containing six video clips from WildEarth-SafariLive, a locally produced wildlife series. Each video clip takes approximately 15–20 min to complete and allows users to explore the African wilderness while listening to calming and atmospheric sounds. In the given context, the control represents an

enhancement of standard care since most adolescents with depression would not usually receive any intervention or active symptom monitoring and referral.

### Intervention arm: Kuamsha programme

The intervention arm will receive six modules of BA therapy via a smartphone application (the Kuamsha app) supported by trained peer mentors, implemented over 10 weeks. The app and phone calls from the peer mentors together comprise the Kuamsha programme.

The Kuamsha app is primarily an interactive narrative game consisting of six tailored modules (sessions) containing BA's core principles integrated into the gamified story content format.[38] Each module takes approximately 15–20 mins to complete, and they cover topics such as identifying and engaging in meaningful activities and using strategies to overcome barriers, for example, using problem-solving, effective communication, getting enough sleep and disengaging from rumination. Each module is followed by a homework activity where the participant is encouraged to reflect on the BA principles outlined in each module and think about ways to apply the principles to their own lives. Participants will be asked to report how often they completed the homework activities and their mood as they were doing these activities. The app includes game design elements to stimulate motivation and performance, including character personification, in-app points and reminders/notifications. Example mock-ups of the Kuamsha app and a summary of each module are shown in online supplemental tables 2–4.

The Kuamsha app will be supplemented by weekly phone calls (15 mins per module) from trained peer mentors. There will be seven calls in total, including one introductory phone call and six calls to cover module content. Peer mentors will attempt to reach participants by calling up to five times per week. The role of the peer mentors is mainly to support adherence and compliance with the app, troubleshoot problems related to the use of the app and assist with the implementation of the homework activities. They will be trained not to provide additional advice or counselling and conduct their calls according to a predetermined checklist of activities to help ensure fidelity. Peer mentors will be Xitsonga-speaking students or recent graduates from the department of psychology or social work at an accredited South African university. The trial psychologist will train and supervise them according to the training manual developed specifically for the intervention.

### Primary objectives and outcome measures
#### Feasibility and acceptability of the intervention.

A mixed methods approach will be adopted to establish feasibility and acceptability.

▶ Feasibility will be assessed by collecting data on the following:
1. Recruitment (enrolment rate of eligible participants) and retention in the trial at the end of the intervention period (11 weeks).
2. Feasibility of testing procedures and data collection methods, including assessment of completion rates.
3. Treatment adherence rates, where adherence is defined as having opened at least four out of six of the app modules and as having completed three out of six phone calls with the peer mentor (excluding the introductory call). We will complement the treatment adherence rates with engagement metrics collected via the app (number of times participants logged into the app, number of modules opened and completed, total time spent on the app, number of weekly activities set to do and number of times the participant completed the weekly activities).

▶ Acceptability of the intervention and study procedures will be assessed via:
1. An acceptability questionnaire conducted at the end of intervention assessment (week 11) with all participants. The questionnaire consists of three measures: Acceptability of Intervention Measure, Intervention Appropriateness Measure and Feasibility of Intervention Measure.[39] Each measure consists of four items. The total score ranges from 1 to 5 and is calculated by averaging response scores to the response categories. We will calculate an average score for each of the measures. We will inquire about the acceptability of the app using these three measures (to both the intervention and control groups) and the acceptability of the peer mentor programme (to the intervention group only).
2. In-depth interviews with a subsample of participants. We estimate that interviews with 20 participants will provide a sufficient range of experiences and perspectives to reach data saturation. This subsample will contain participants from the treatment and control arms and will be stratified based on high versus low app engagement.

▶ Fidelity of delivery of the intervention will be assessed by collecting data on adherence and competence of trained peer mentors.
1. Adherence of the peer mentors is defined as the number of sessions that meet at least 90% of the criteria for adherence according to the training protocol. Independent raters will listen to a random sample (10%) of recordings of peer mentors' phone calls with participants and assess them against the training protocol.
2. Competence of the peer mentors will be expressed as a percentage based on their competency assessment test. This test will include a written test and observation of skills through role-playing to assess knowledge, attitudes and practices. Tests will be scored by the trial psychologist using a predetermined scoring system.

**Table 1** Trial progression criteria

| Criteria | Green | Amber | Red |
|---|---|---|---|
| Enrolment (recruitment) rate of eligible participants | ≥60% | <60%, ≥40% | <40% |
| Retention in the trial at 11 weeks | ≥90% | <90%, ≥50% | <50% |
| Proportion of participants that open at least four out of six of the app modules | ≥70% | <70%, ≥50% | <50% |
| Proportion of participants that have three out of six phone calls with the peer mentor | ≥70% | <70%, ≥50% | <50% |

## Trial progression criteria

Feasibility of the intervention and trial procedures and progression criteria for a definitive Randomised Controlled Trial (RCT) are given in table 1. These will be based on a traffic light system of green (continue to RCT), amber (make modifications to trial procedures before embarking on a definitive RCT) and red (a definitive RCT is unlikely to be feasible).[40]

## Signals of initial efficacy on depressive symptoms.

This will be assessed using the PHQ-A Score at the end of intervention assessment (week 11). The PHQ-A is a widely used and well-established measure of adolescent depressive symptoms over the past 2 weeks.[41] The PHQ-A will be administered in Xitsonga. While this measure has not been validated in Xitsonga, it showed good psychometric properties with a sample of adolescents and young adults in South Africa and Kenya[42–44] and has been validated in various other South African languages, which also form part of the 'Nguni' language group, including isiXhosa,[45] seTswana,[42] isiZulu and seSotho.[43] The questionnaire consists of nine items. The PHQ-A total score ranges from 0 to 27 and is calculated by assigning scores to the response categories (0=not at all, 1=several days, 2=more than half the days, 3-nearly every day) and summing the score for each of the items. Higher scores indicate greater severity of depression. PHQ-A scores of 5, 10, 15 and 20 represent mild, moderate, moderately severe and severe depression, respectively.[41] We will use this scale to assess depressive symptoms at screening, week 5, week 11 and week 24. Participants will also be asked to complete this scale at week 2.5 and week 7.5 as part of the symptom monitoring. More information on how we will analyse this data and trial progression criteria can be found in the data analysis section.

## Secondary objectives and outcome measures

As part of our secondary objectives, we will pilot locally adapted measures of mental health, social–affective and cognitive processing, risky behaviours and socioeconomic outcomes. The overall aim of implementing these measures is to assess their acceptability, feasibility and variation, and use these descriptive data to inform the development of a future larger trial. More details on how we will analyse these secondary outcomes can be found in the SAP in the online supplemental material 1.

## Mental health outcomes

Mental health outcomes include the following scales Generalised Anxiety Disorder[46]; Connor-Davidson Resilience Scale[47]; Brief Rumination Response Scale[48]; Warwick-Edinburgh Mental Wellbeing Scale[49]; Behavioural Activation for Depression Scale.[50] Participants will be asked to complete these scales at baseline (week 0), midintervention (week 5), end of intervention (week 11) and follow-up (week 24). See table 2 for details.

## Social–affective and cognitive processing

Social–affective and cognitive processing include five computerised cognitive tasks measuring affective set-shifting, affective working memory, risk-seeking preferences, emotion recognition and abstract reasoning. The tasks include a modified version of the Wisconsin Card Sorting Task[51]; the Backward Digit Span Task[52]; the Balloon Analogue Risk Task[53]; an Emotion Recognition Task[54]; and the Matrix Reasoning Item Bank Task.[55] Participants will be asked to complete these tasks at baseline (week 0) and end of intervention (week 11).

## Risky behaviours

The questions on sexual behaviours include a subset of items from the Wits Reproductive Health and HIV Institute (Wits RHI) sexual behaviour survey.[56] We will measure substance use using the Alcohol, Smoking and Substance Involvement Screening Test for Youth.[57] We will also ask about delinquency, gambling and peer influence in participants' engagement with risky behaviours. Participants will be asked to complete these tasks at baseline (week 0) and end of intervention (week 11).

## Socioeconomics

We will measure economic preferences (time preference, risk preference and loss aversion) using three incentivised tasks.[58 59] In addition, we will measure a range of socioeconomic outcomes, including measures of time use, human capital investment and spending. Participants will be asked to complete these tasks at baseline (week 0) and end of intervention (week 11).

## Recruitment and participant timeline

Participants will be recruited through a two-stage recruitment process. The first stage of recruitment (phase 1) will consist of a screening survey conducted in schools to identify adolescents with symptoms of mild to moderately severe depression. The second stage of recruitment (phase 2) will be done with adolescents who score

**Table 2** Schedule of enrolment, interventions and assessments for participants

| Assessment | Study period | | | | | | | |
| --- | --- | --- | --- | --- | --- | --- | --- | --- |
| | Enrolment | Allocation | Post allocation | | | | | Close-out |
| Time point (in weeks) | $-t_1$ | 0 | 1 | 2.5 | 5 | 7.5 | 11 | 24 |
| Enrolment | | | | | | | | |
| Eligibility screen (school survey) | X | | | | | | | |
| Informed consent | X | | | | | | | |
| Allocation | | X | | | | | | |
| Phone delivery | | X | | | | | | |
| Intervention | | | | | | | | |
| Intervention arm: Kuamsha app and peer mentor calls | | | ●━━━━━━━━━━━━━● | | | | | |
| Control arm: Kuchunguza app | | | ●━━━━━━━━━━━━━● | | | | | |
| Assessments | | | | | | | | |
| School survey: PHQ-A, sociodemographic questions and eligibility checklist | X | | | | | | | |
| Baseline assessment: demographic questionnaire, GAD-7, CD-RISC, RRS, WEMWBS, BADS, five social–affective and cognition tasks, risky behaviours and socioeconomic outcomes | X | | | | | | | |
| Symptom monitoring: PHQ-A | | | | X | | X | | |
| Midintervention assessment: PHQ-A, GAD-7, RRS, WEMWBS short form, BADS (activation and social impairment subscales) | | | | | X | | | |
| End of intervention assessment: same as baseline+PHQ-A+acceptability questionnaire and in-depth interviews with subsample | | | | | | | X | |
| Follow-up assessment: PHQ-A+GAD-7, RRS, BADS (activation subscale), WEMWBS short form | | | | | | | | X |

BADS, Behavioural Activation for Depression Scale; CD-RISC, Connor-Davidson Resilience Scale; GAD-7, Generalised Anxiety Disorder; PHQ-A, Patient Health Questionnaire Adolescent Version; RRS, Rumination Response Scale; WEMWBS, Warwick-Edinburgh Mental Wellbeing Scale.

between 5 and 19 on the PHQ-A in the school survey and who also meet the other eligibility criteria.

The pilot trial consists of an 11-week intervention treatment phase, and participants will be followed up for a further 13 weeks after completing the intervention (ie, a total of 24 weeks). Primary and secondary outcomes will be assessed at the end of intervention assessment (week 11). We will use different methods to assess these outcomes. Specifically, we will ask about mental health (including PHQ-A) and risky behaviours using Audio Computer-Assisted Self-Interviewing software. Under this approach, participants will listen to prerecorded survey questions through headphones and select their responses on a tablet computer. The behavioural tasks will be performed individually by participants on a tablet computer, in a quiet location under the instruction of a trained fieldworker. All app engagement metrics (including treatment adherence) will be captured on an online database automatically as participants engage with the app. The acceptability questionnaire will be sent via

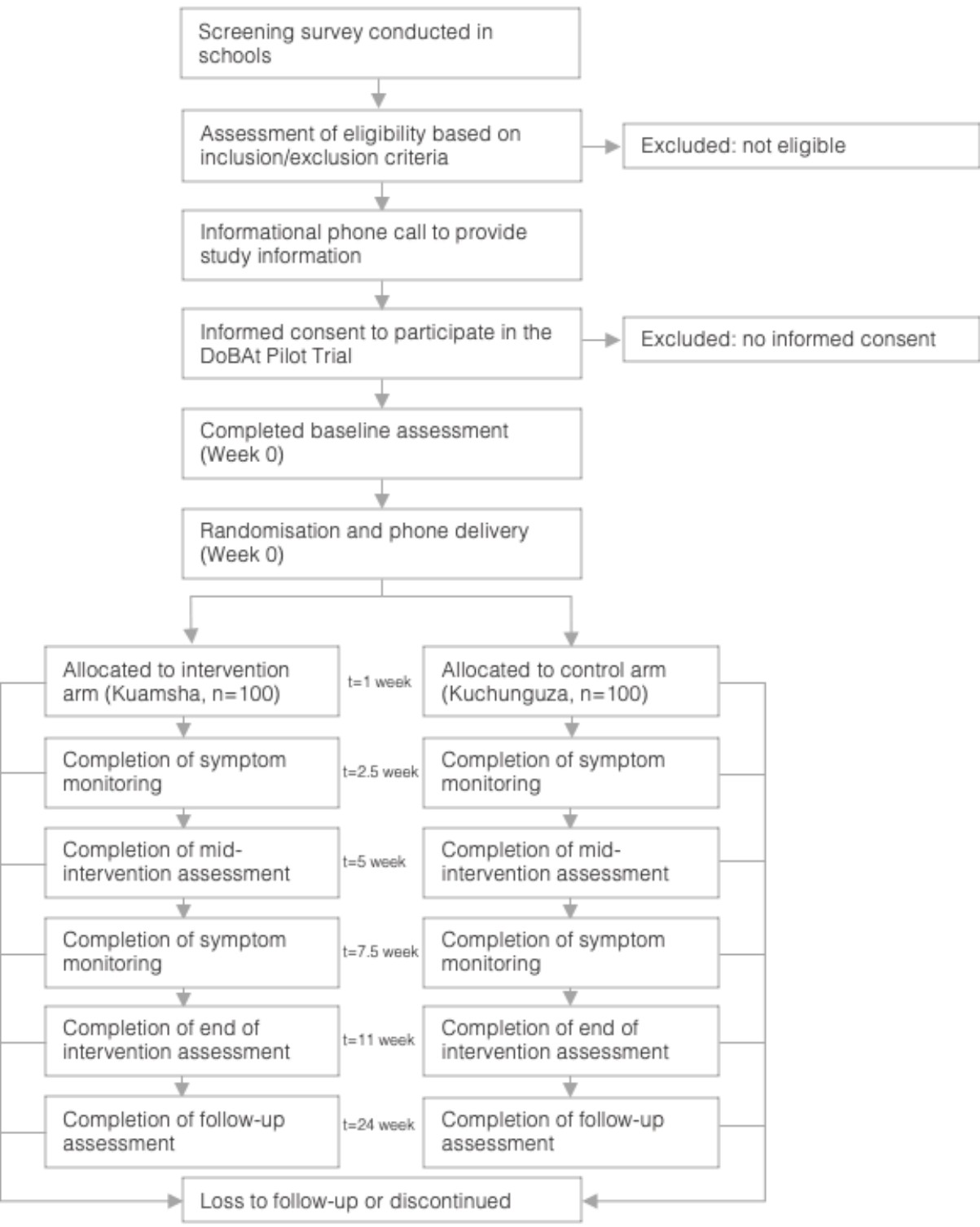

**Figure 1** CONSORT Flow Diagram for the DoBAt Study.

text message to the study phones. The socioeconomic outcomes will be assessed by blinded outcomes assessors. Table 2 indicates the time schedule of enrolment, interventions and assessments for participants, and figure 1 shows the flow of participants.

**Allocation and blinding**

Participants will be randomly assigned to the intervention or control arm with a 1:1 allocation using a computerised minimisation algorithm, balanced by sex (male or female) and severity of depressive symptoms (<10 or ≥10 on the PHQ-A). The minimisation algorithm was generated by

the Centre for Healthcare Randomised Trials (CHaRT) at the University of Aberdeen. Participants will be allocated using CHaRT's online software and the trial manager will oversee the enrolment of participants as per assignment.

For practical and risk management reasons, the trial manager, trial psychologist, peer mentors and field-workers delivering the phones to participants and receiving calls to troubleshoot problems with the app/phone (ie, 'phones team') will not be blinded. However, all fieldworkers conducting outcomes assessments (ie, the 'assessments team') will be blinded to participants' allocation status. Any breaches in blinding will be documented, and we will ensure to change the fieldworker conducting subsequent assessments on a particular adolescent where unblinding has occurred.

### Analysis and statistical methods

Quantitative data will be analysed using Stata V.14.0,[60] R[61] and other appropriate statistical analysis packages under the direction of the trial statistician.

Participants will be analysed in the groups to which they are randomly assigned, regardless of deviation from the protocol or treatment received (intention to treat population) and the data analyst will be blind to arm allocation. We will use descriptives to explore patterns in the data, followed by inferential statistics involving univariate and multivariable models. A two-tailed p value<0.05 will be considered statistically significant in the inferential analyses.

### Primary objectives

Our first coprimary objective (feasibility and acceptability) will be assessed using a mixed methods approach. We will compute appropriate summary statistics (eg, proportions, means, SDs, etc) for each quantitative outcome and evaluate the feasibility of the trial based on the predefined progression criteria discussed earlier.

Qualitative interviews will be audio-recorded, transcribed verbatim and translated into English. Transcripts will be analysed using thematic analysis. We will follow Braun and Clarke's six phases of analysis (ie, becoming familiar with the data, generating initial codes, searching for themes, reviewing themes, defining and naming themes and producing the report).[62] Coding will be done by two independent researchers. NVivo V.10, a computer programme that aids in the sorting and management of qualitative data, will be used to facilitate the analysis.

Our second coprimary objective (initial efficacy on depressive symptoms) will be based on the PHQ-A Score (continuous) at 11 weeks. Outcomes will be compared between intervention and control groups using a linear regression model and adjusting for covariates (PHQ Score measured at baseline, sex, age, depression severity and household asset index). As a secondary analysis, we will make use of the repeated measurements of the PHQ-A throughout the trial (up to six times per individual) to evaluate the treatment effects over time. The details of this analysis will be documented in the SAP.

### Secondary objectives

The descriptive data of the secondary outcomes will include possible floor and ceiling effects, accuracy, duration, latency and efficiency. We will perform standard psychometric tests to examine whether the instruments perform well in the study context (eg, reliability, validity, acceptability) and calculate bivariate correlations across variables of interest. We anticipate that some secondary analyses will be included in companion papers rather than the main paper. The details of this analysis will be documented in a separate analysis plan.

### Sample size considerations

The sample size was calculated on the basis of one of our coprimary objectives, which aims to provide preliminary evidence of signals of efficacy (direction and magnitude) of any effects of the intervention on depressive symptoms (measured by the PHQ-A) among adolescents in the intervention arm compared with the control group.

Statistical power was calculated to detect differences between two independent groups, in a two-sided test with an α of 0.05 and a power of 1–β=0.80, for an effect size of 0.45. This effect size is based on findings from previous studies investigating digital psychological interventions, with effect size (Cohen's d) ranging from 0.24 to 0.57.[29 63 64] Given these findings, we chose a small-to-medium effect size of 0.45 to account for the limited number of studies conducted with our target population and in our study setting. We allowed for 25% attrition, based on a previous study with adolescents in the Agincourt setting.[65] Given these assumptions, we aimed to recruit 200 participants at baseline. We calculated the required sample size using G*Power software, V.3.1.9.3.[66]

### Trial status

Enrolment of the first participant occurred on 25 November 2021. The trial is currently ongoing, and we expect to finish data collection in January 2023.

## ETHICS AND DISSEMINATION
### Research ethics approval

Ethical approval for this study has been obtained from the University of the Witwatersrand Human Research Ethics Committee (MED20-05-011), Ehlanzeni District and Mpumalanga Provincial Departments of Health and Education and the Oxford Tropical Research Ethics Committee (OxTREC 34-20). This trial was registered with the South African National Clinical Trials Registry (DOH-27-112020-5741) and the Pan African Clinical Trials Registry (PACTR202206574814636) in November 2020. Further details on Trial Registration can be found in online supplemental table 5.

### Informed consent

At both stages of recruitment, we will obtain informed consent from participants aged 18 or over and informed assent and parental/guardian consent from participants

younger than 18. All information sheets will be available in Xitsonga and English, and assent/consent will be obtained by trained and supervised bilingual fieldworkers.

## Confidentiality and management of participant data

The confidentiality of participants' data and information will be respected and maintained by all study staff. Staff members will be trained accordingly and required to sign a non-disclosure agreement. A unique participant identification number will be used to link study data and information. All electronic data will be stored in a secure, protected and access-controlled database at the Agincourt data centre. Paper-based documentation will be stored in a locked cabinet and only accessible to authorised staff. Confidentiality may be broken in instances of immediate harm to self or others, as detailed in consent and assent forms.

## Harms/anticipated risks

A comprehensive risk management protocol has been developed to ensure the safety of participants in the trial. Participants with severe depression, suicidal ideation and other risks will receive a telephonic and/or in-person assessment by a member of the risk management team. Subsequently, appropriate referrals and linkage to care will be made using established networks with local providers. Details of the independent data and safety monitoring board (DSMB) and trial steering committee are provided in online supplemental table 1. Serious adverse events will be reported to the DSMB chair within 48 hours and to both ethics committees within 7 days. Summary tables of all adverse events will be sent to the DSMB on a quarterly basis and to the ethics committees on an annual basis.

## Publication and dissemination

Results of the trial will be communicated to participants, the public, researchers, healthcare professionals and policy-makers. We will seek the involvement of adolescents and members of the public in the development of appropriate methods to disseminate study findings. Policy-makers and other key stakeholders within provincial and national departments of health, education and social development, as well as relevant local NGOs, will be engaged in dialogue and supplied with a technical brief to convey the results of the trial and their implications for policy and practice. Trial results will be published as soon as possible after completion, and ensuing publications will be made open access. Results will also be presented at relevant national and international conferences. Authorship will be determined in accordance with the ICMJE guidelines, and other contributors will be acknowledged.

**Author affiliations**

[1] MRC/Wits Rural Public Health and Health Transitions Research Unit (Agincourt), School of Public Health, Faculty of Health Sciences, University of the Witwatersrand, Johannesburg, South Africa
[2] Department of Psychiatry, University of Oxford, Oxford, UK
[3] Center for Community Based Research, Human Sciences Research Council, Pietermaritzburg, South Africa
[4] MRC/Wits Developmental Pathways for Health Research Unit, University of the Witwatersrand, Johannesburg, South Africa
[5] Mood Disorders Centre, Department of Psychology, University of Exeter, Exeter, UK
[6] Department of Psychiatry and Biobehavioral Sciences, University of California Los Angeles (UCLA), Los Angeles, California, USA
[7] Department of Psychology, University of California Los Angeles (UCLA), Los Angeles, California, USA
[8] Psychology, University of Limpopo, Sovenga, Limpopo, South Africa
[9] Alan J Flisher Centre for Public Mental Health, Department of Psychiatry and Mental Health, University of Cape Town, Rondebosch, South Africa
[10] Centre for Global Mental Health, Institute of Psychiatry, Psychology and Neuroscience, King's College London, London, UK
[11] Blavatnik School of Government, University of Oxford, Oxford, UK
[12] Department of Clinical, Educational and Health Psychology, University College London, London, UK
[13] Department of Psychology, University of Cambridge, Cambridge, UK
[14] Department of Economics, University of Exeter, Exeter, UK
[15] Division of Epidemiology and Biostatistics, School of Public Health, Faculty of Health Sciences, University of the Witwatersrand, Johannesburg, South Africa
[16] Department of Psychology, University of Limpopo, Polokwane, South Africa
[17] Southern Centre for Inequality Studies, University of the Witwatersrand, Johannesburg, South Africa
[18] Umeå Centre for Global Health Research, Division of Epidemiology and Global Health, Department of Public Health and Clinical Medicine, Umeå Universitet, Umea, Sweden

**Acknowledgements** We would like to acknowledge the field supervisors Meriam Meritze, Princess Makhubela and Nokthula Mayindi, as well as all the fieldworkers, registered counsellor, Tinyiko Mafumo and peer mentors for their hard work and dedication. We would also like to thank all the adolescents who were involved in the formative work and participatory workshops that were integral to the design and development of the intervention, as well as the educators, healthcare workers and non-governmental organisations that provided valuable input into the trial design, particularly regarding recruitment and risk management strategies.

**Contributors** AS, KK, ST, MC, HAO, S-JB, CL, TS, KO, JRP, EJK, BDM, MM, FXG-O, IV and AvH made substantial contributions to the conception and design of the study. BDM, JRP, MD, SLF, GC and DM were responsible for programming study measures and training field teams. ZM and TB trained and supervised peer mentors. JRP, KO and EM drafted the statistical analysis plan. The first draft of the manuscript was written by BDM and JRP. All authors reviewed the manuscript, and gave the final approval of the version to be published. BDM submitted the manuscript.

**Funding** Research reported in this publication was supported by the South African Medical Research Council with funds received from the South African National Department of Health and the UK Medical Research Council, with funds received from the UK Government's Newton Fund (grant number MR/S008748/1). It also received financial support from the NIHR Oxford Health Biomedical Research Centre (grant number: N/A). The study was conducted within the MRC/Wits Rural Public Health and Health Transitions Research Unit and Agincourt Health and socio-Demographic Surveillance System, a node of the South African Population Research Infrastructure Network (SAPRIN), supported by the Department of Science and Innovation, the University of the Witwatersrand and the Medical Research Council, South Africa (grant number: N/A). S-JB is funded by Wellcome (grant number WT107496/Z/15/Z), the MRC-UK, the Jacobs Foundation, the Wellspring Foundation and the University of Cambridge (grant number: N/A). JP is funded by Fundación Rafael del Pino and received funding support from MRC UK and the Wellspring Philanthropic Fund (grant number: N/A). EJK received funding support from a UKRI-GCRF Impact Acceleration Award (grant number: N/A). Study funders played no role in the study design, collection, management, analysis or interpretation of data, the writing of this report, or the decision to submit it for publication.

**Competing interests** None declared.

**Patient and public involvement** Patients and/or the public were involved in the design, or conduct, or reporting, or dissemination plans of this research. Refer to the Methods and analysis section for further details.

**Patient consent for publication** Not applicable.

**Provenance and peer review** Not commissioned; externally peer reviewed.

**ORCID iDs**
Bianca D Moffett http://orcid.org/0000-0002-8887-1374
Julia R Pozuelo http://orcid.org/0000-0002-3058-0371
Alastair van Heerden http://orcid.org/0000-0003-2530-6885
Tholene Sodi http://orcid.org/0000-0001-7592-5145
Crick Lund http://orcid.org/0000-0002-5159-8220
Kate Orkin http://orcid.org/0000-0002-3447-9094
Emma J Kilford http://orcid.org/0000-0002-0360-3769
Sarah-Jayne Blakemore http://orcid.org/0000-0002-1690-2805
Gabriele Chierchia http://orcid.org/0000-0002-5623-4573
F Xavier Gómez-Olivé http://orcid.org/0000-0002-4718-9336
Imraan Valodia http://orcid.org/0000-0002-5607-6595
Stephen Tollman http://orcid.org/0000-0003-0744-7588
Kathleen Kahn http://orcid.org/0000-0003-3339-3931
Alan Stein http://orcid.org/0000-0001-8207-2822

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
