## [Reviewer comments · BMJ Open]

ARTICLE DETAILS

TITLE (PROVISIONAL)	Digital delivery of Behavioural Activation therapy to overcome depression and facilitate social and economic transitions of adolescents in South Africa (the DoBA study): protocol for a pilot randomised controlled trial
AUTHORS	Moffett, Bianca; Pozuelo, Julia; van Heerden, Alastair; O'Mahen, Heather; Craske, Michelle; Sodi, Tholene; Lund, Crick; Orkin, Kate; Kilford, Emma J; Blakemore, Sarah-Jayne; Mahmud, Mahreen; Musenge, Eustasius; Davis, Meghan; Makhanya, Zamakhanya; Baloyi, Tlangelani; Mahlangu, Daniel; Chierchia, Gabriele; Fielmann, Sophie L.; Gómez-Olivé, F. Xavier; Valodia, Imraan; Tollman, Stephen; Kahn, Kathleen; Stein, Alan

VERSION 1 – REVIEW

REVIEWER	Mueller-Weinitschke, Claudia University of Freiburg, Department of Rehabilitation Psychology and Psychotherapy, Institute of Psychology
REVIEW RETURNED	07-Jul-2022

GENERAL COMMENTS	This manuscript presents a study protocol for a pilot randomised controlled trial on the effects of digitally delivered behavioural activation therapy on depression and other secondary outcomes in adolescents in South Africa. The overall design is reasonable, the manuscript well written. However, the following points should be considered: 1. In the trial registration you state that Enrolment of the first participant occurred on the 25th November 2021. Please state in the manuscript, that the trial is currently ongoing.2. Title: You state that “overcoming depression” is the aim of the study. This is an ambitious goal not reflected in the research question, as no depression diagnoses are made. Rather, the aim is to reduce depressive symptoms.3. Abstract (p.4, l. 10): Results on the feasibility, acceptability, and effectiveness of digitally delivered BA interventions already exists (cf. review by Huguet et al. 2018). These findings can certainly be expanded and corroborated. Please clarify.4. Introduction: The selection of secondary outcomes is not yet sufficiently justified in the introduction. What added value do the authors expect from the recording of social-affective, cognitive, and socioeconomic variables? This should be further elaborated.5. P. 6, l. 54: It is the function of a study protocol to describe the planned research project in a detailed and replicable manner. This is not possible due to the outsourcing of the statistical analyses to a SAP. Please provide the statistical analysis plan.6. P. 10, l. 6: “Independent raters will listen to a random sample of recordings of Peer Mentors’
--

	phone calls with participants and assess them against the training protocol." How many recordings will be assessed? 7. P. 10: Could you elaborate why no adverse events, or acceptability outcomes are considered in the trial progression criteria. 8. P. 10, l. 38: Will the PHQ-A be used in Xitsonga or in English? Is the used version of the PHQ-A already validated in the language? This question also applies to the other adapted measures. 9. P. 11, l. 51: "Primary outcomes will be assessed at the end of intervention assessment (week 11) by blinded outcomes assessors." This statement is misleading, as the participants themselves are the outcome assessors and assumably not blinded to the condition. 10. P. 12, Table 1: Please add an overview of the abbreviations used in the table. 11. P. 12, l. 54: Which researcher is responsible for randomization? Is this person independent? 12. P. 13, l. 44: No information about the handling of missing data is provided. What method is used to account for missing values in your regression analysis? 13. P. 14, 2.11: It is not clear to which outcome of your study this power analysis refers. Since the primary outcome is feasibility, the power analysis should also refer to this outcome. Is this the case? Or was the power analysis performed for the secondary outcome of depression severity as suggested by the cited articles? Please clarify. 14. This protocol lacks a discussion. Please elaborate on the strengths and limitations of your study, taking into account its contribution to the current state of research on digital BA interventions. 15. Supplemental materials: As per Item No. 32 of the SPIRIT Checklist please add the informed consent materials in the Supplement.
--	---

REVIEWER	Foulds, James A Department of Psychological Medicine, University of Otago
REVIEW RETURNED	21-Oct-2022

GENERAL COMMENTS	This is a clinical trial protocol for a peer-supported behavioural activation intervention delivered via mobile app to depressed young people in South Africa. This is primarily a pilot feasibility and acceptability study, which will presumably be extended in a later, larger trial depending on the findings. The protocol clearly articulates the objectives of the study and its proposed conduct. Page 6 line 24: what is known about the characteristics of online treatments for adolescent mental health problems that produce good adherence? How will those characteristics that promote high adherence be integrated into the proposed trial? Section 2.2: this may go without saying, but I presume there is good mobile phone service coverage throughout the districts where the participants will be living? One minor comment- suggest to include reference for PHQ score interpretation (page 7 line 50).
--

REVIEWER	Bulgheroni, Maria
-----------------	-------------------

	Ab.Acus srl, Milan, IT, R&D
REVIEW RETURNED	23-Oct-2022

GENERAL COMMENTS	The study is well designed and clear and repeatable (give the availability of similar digital tools). I have a concern referring the relationship between the design and the objectives that is not fully clear to me. The primary objective of the study is "to determine feasibility and acceptability of a digitally delivered BA therapy intervention ...". I think that it is not clear enough what "digitally delivered BA intervention" means. Looking at the control and the intervention group, the main difference is not only the used app but also the support of trained Peer Mentors. The phone support plays a key role in the administration and follow up of the intervention. So, if this is the case (i.e. app+phone to be evaluated) it is important to clarify that the digital intervention consists not only of a digital application but also of a key phone support since the early phases of the manuscript. While, if the study aims at focusing on the digital app alone, it is needed to supply some form of phone support also to the contro group. Please, clarify this issue in the manuscript. A second point refers to the method for assessing the digital competencies needed to run the digital app (inclusion criterion 3: is it only a matter of language knowledge of digital alphabetization? you are going to deliver them a new phone that they are not used to use and this may become a barrier; and exclusion criterion 6: what do you mean by capacity to use the app?) A not homogeneous digital capability of the participants may impact on the reliability of the results of the study. Please, clarify the way your are going to assess digital capabilities at the recruitment time.
---

VERSION 1 – AUTHOR RESPONSE

Reviewer 1: *Ms Claudia Mueller-Weinitschke, University of Freiburg*

This manuscript presents a study protocol for a pilot randomised controlled trial on the effects of digitally delivered behavioural activation therapy on depression and other secondary outcomes in adolescents in South Africa. The overall design is reasonable, the manuscript well written. However, the following points should be considered:

1. In the trial registration you state that Enrolment of the first participant occurred on the 25th November 2021. Please state in the manuscript that the trial is currently ongoing.

Authors' response

We have added this statement to the revised manuscript as follows (page 12):

2.12 Trial status

Enrolment of the first participant occurred on the 25th of November 2021. The trial is currently ongoing, and we expect to finish data collection in January 2023.

2. Title: You state that “overcoming depression” is the aim of the study. This is an ambitious goal not reflected in the research question, as no depression diagnoses are made. Rather, the aim is to reduce depressive symptoms.

Authors’ response

Thank you for this suggestion. The title of the study was agreed upon by the Trial Management Group at the time of the grant review by the Medical Research Council and reflects the aims of the broader study, which is to develop and evaluate a digital intervention to overcome depression. In low-resource settings, there is a severe shortage of mental health professionals and it is seldom feasible to have a mental health professional make a clinical diagnosis of depression using a standardised diagnostic instrument as would be the gold-standard in high-resource settings. Therefore, depression treatment trials in low-resource settings frequently rely on the use of depression screening tools such as the PHQ-9 to detect depression, as was done, for example by Professor Vikram Patel et al in the Health Activity Program, published in the Lancet in 2017 (Patel et al., 2017).

We do agree with Reviewer 1 that the use of a depression screening tool as opposed to a clinical diagnosis of depression should be noted as a study limitation, and have included this in the list of ‘Strengths and Limitations’ in the manuscript (page 2-3). Furthermore, we are committed to acknowledging this limitation in the manner in which we report our study findings, as reflected in the way we have framed the co-primary objectives of our study (page 4):

The co-primary objectives of this pilot study are: 1) to determine the feasibility and acceptability of a digitally delivered Behavioural Activation therapy intervention for adolescents living with depression in the Bushbuckridge sub-district of Mpumalanga province, South Africa; and 2) to provide preliminary evidence on any signals of initial efficacy (direction and magnitude) of any effects of the intervention on depressive symptoms amongst adolescents in the intervention arm compared to control.

3. Abstract (p.4, l. 10): Results on the feasibility, acceptability, and effectiveness of digitally delivered BA interventions already exists (cf. review by Huguet et al. 2018). These findings can certainly be expanded and corroborated. Please clarify.

Authors’ response

We thank Reviewer 1 for this suggestion. We have modified the abstract (page 2) as follows:

Whilst digital delivery of Behavioural Activation presents a promising solution, its feasibility, acceptability, and effectiveness amongst adolescents in an African context remains to be shown.

Further, we have expanded the discussion regarding these findings in the introduction of the revised manuscript (page 4; see references at the end of this document):

Despite the many potential advantages of digital mental health interventions, studies that have evaluated their effectiveness among adolescents have yielded mixed results (Eilert et al., 2022; Lehtimaki et al., 2021). Most of the evidence on the efficacy of digitally-delivered BA

comes from studies conducted in high-income countries (Huguet et al., 2018), and it is, therefore, an open question whether it is possible to generalise these findings to an African context where conditions and resources differ vastly (Lehtimaki et al., 2021; Naslund et al., 2017). Furthermore, studies of online self-help interventions amongst adults in LMICs showed high attrition rates (Naslund et al., 2017), and adherence to mental health apps in all contexts is often low (Linardon & Fuller-Tyszkiewicz, 2020). While some commercial smartphone apps attract more users, many have not been rigorously evaluated and show little fidelity to evidence-based treatments (Schueller & Torous, 2020)

4. Introduction: The selection of secondary outcomes is not yet sufficiently justified in the introduction. What added value do the authors expect from the recording of social-affective, cognitive, and socioeconomic variables? This should be further elaborated.

Authors' response

This decision was based on the following evidence:

1. Depression has been associated with worse performance in social-affective cognitive tasks; and social cognitive deficits, in turn, have been shown to play a role in predicting treatment response and in determining a range of social and economic outcomes (Diamond, 2013; Snyder, 2013). This evidence was described on page 3 (second paragraph of the introduction). Given this, we believed it was important to include these tasks to assess their acceptability, feasibility and variation, and use these descriptive data to inform the development of a further larger trial.
2. Accumulating evidence shows that socioeconomic variables such as poverty and depression interact in a vicious cycle, with a higher prevalence of depression in conditions of poverty (Ridley et al., 2020). We, therefore, included these variables as possible covariates in our primary inference. Further, we will also analyse the treatment effects of the intervention on these outcomes, but the analysis is exploratory, to inform future work, rather than confirmatory. We will focus on the direction and magnitude of the effects of the intervention.

We have added a sentence explaining the rationale for this in the revised manuscript as follows (page 9):

The overall aim of implementing these measures is to assess their acceptability, feasibility, and variation, and use these descriptive data to inform the development of a future larger trial. More details on how we will analyse these secondary outcomes can be found in the statistical analysis plan in the supplemental materials.

5. P. 6, l. 54: It is the function of a study protocol to describe the planned research project in a detailed and replicable manner. This is not possible due to the outsourcing of the statistical analyses to a SAP. Please provide the statistical analysis plan.

Authors' response

We thank Reviewer 1 for raising this important point. We have now added the statistical analysis plan (SAP) as part of the supplemental materials.

6. P. 10, l. 6: "Independent raters will listen to a random sample of recordings of Peer Mentors' phone calls with participants and assess them against the training protocol." How many recordings will be assessed?

Authors' response

Thank you for requesting this clarification. Independent raters will assess 10% of the Peer Mentors' phone calls with participants. We have clarified this in the protocol paper (page 8) as follows:

Independent raters will listen to a random sample (10%) of recordings of Peer Mentors' phone calls with participants and assess them against the training protocol.

7. P. 10: Could you elaborate why no adverse events, or acceptability outcomes are considered in the trial progression criteria.

Authors' response

We thank Reviewer 1 for this important question. Our trial progression criteria were decided on by the Trial Management Team and reviewed as part of the Statistical Analysis Plan by the Trial Steering Committee and the Data and Safety Monitoring Board. We considered various options for progression criteria, including the criteria the reviewer mentions, but decided to base our choice of progression criteria on this seminal paper by Avery et al. (2017). In line with the Statistical Analysis Plan, we will however still report on both acceptability outcomes and adverse events from the trial.

8. P. 10, l. 38: Will the PHQ-A be used in Xitsonga or in English? Is the used version of the PHQ-A already validated in the language? This question also applies to the other adapted measures.

Authors' response

Thank you for raising this important point. Regarding our primary outcomes, we have clarified this in the manuscript as follows (page 8):

The PHQ-A will be administered in Xitsonga. Whilst this measure has not been specifically validated in Xitsonga, it showed good psychometric properties with a sample of adolescents and young adults in South Africa and Kenya (Bhana et al., 2015; Cholera et al., 2014; Osborn et al., 2019) and has been validated in various other South African languages which also form part of the 'Nguni' language group, including isiXhosa (Marlow et al., 2022), seTswana (Bhana et al., 2015), isiZulu and seSotho (Cholera et al., 2014)

Regarding our secondary outcomes, further details on their use and validation is provided in the Statistical Analysis Plan which we have now included in the Supplemental Material. Whilst these measures have not yet been specifically validated in Xitsonga, including these measures in our pilot study will allow us to perform standard psychometric tests to examine how well they perform in the study context.

9. P. 11, l. 51: "Primary outcomes will be assessed at the end of intervention assessment (week 11) by blinded outcomes assessors." This statement is misleading, as the participants themselves are the outcomes assessors and assumably not blinded to the condition.

Authors' response

We have clarified this point in the revised manuscript as follows (page 10):

Primary and secondary outcomes will be assessed at the end of intervention assessment (week 11). We will use different methods to assess these outcomes. Specifically, we will ask about mental health (including PHQ-A) and risky behaviours using Audio Computer-Assisted Self-Interviewing (ACASI) software. Under this approach, participants will listen to pre-recorded survey questions through headphones and select their responses on a tablet computer. The behavioural tasks will be performed individually by participants on a tablet computer, in a quiet location under the instruction of a trained fieldworker. All app engagement metrics (including treatment adherence) will be captured on an online database automatically as participants engage with the app. The acceptability questionnaire will be sent via text message to the study phones. The socioeconomic outcomes will be assessed by blinded outcomes assessors.

10. P. 12, Table 1: Please add an overview of the abbreviations used in the table.

Authors' response

We have amended this as suggested (page 11).

11. P. 12, l. 54: Which researcher is responsible for randomization? Is this person independent?

Authors' response

Researchers in the DoBAAt study team do not play any role in the actual randomisation process. This is controlled by the Centre for Healthcare Randomised Trials (CHaRT) at the University of Aberdeen an organisation entirely independent of the DoBAAt study team,. The minimisation algorithm and software that is used was developed by the Centre for Healthcare Randomised Trials (CHaRT) at the University of Aberdeen.

Thus the steps to randomise participants are the following:

- 1) Trial manager logs in to CHaRT software
- 2) Trial manager inputs the following information: Sex, PHQ-A score, and an anonymised participant identifier
- 3) The CHaRT software performs minimisation and sends an email to the Trial manager with the allocation

12. P. 13, l. 44: No information about the handling of missing data is provided. What method is used to account for missing values in your regression analysis?

Authors' response

This information is described in the Statistical Analysis Plan, which is now part of the supplemental materials. In short:

Patterns in missing data will be explored for all analyses. Complete case analyses will be reported in all instances, but in the event of data that appear to be missing not at random, we will also consider the use of imputation methods as appropriate.

13. P. 14, 2.11: It is not clear to which outcome of your study this power analysis refers. Since the primary outcome is feasibility, the power analysis should also refer to this outcome. Is this the case? Or was the power analysis performed for the secondary outcome of depression severity as suggested by the cited articles? Please clarify.

Authors' response

This pilot trial has two primary objectives:

- 1) To determine the feasibility and acceptability of the digital intervention
- 2) To provide preliminary evidence of signals of efficacy (direction and magnitude) of any effects of the intervention on depressive symptoms (measured by the PHQ-A) amongst adolescents in the intervention arm compared to the control group.

The statistical power was calculated on the basis of one of these primary objectives (number 2 above). We have now included this as part of the main text of the manuscript as follows (page 12):

The sample size was calculated on the basis of one of our co-primary objectives which aims to provide preliminary evidence of signals of efficacy (direction and magnitude) of any effects of the intervention on depressive symptoms (measured by the PHQ-A) amongst adolescents in the intervention arm compared to the control group.

14. This protocol lacks a discussion. Please elaborate on the strengths and limitations of your study, taking into account its contribution to the current state of research on digital BA interventions.

Authors' response

The BMJ open authors' guidelines state that discussions are optional. Given that we were already above the word count, we decided to exclude this from the protocol paper but would be happy to add this section if required.

15. Supplemental materials: As per Item No. 32 of the SPIRIT Checklist please add the informed consent materials in the Supplement.

Authors' response

An example of the patient consent form has now been added to the supplemental materials, as suggested by the reviewer and BMJ's editor.

Reviewer: 2; Dr. James A Foulds, Department of Psychological Medicine, University of Otago

This is a clinical trial protocol for a peer-supported behavioural activation intervention delivered via mobile app to depressed young people in South Africa. This is primarily a pilot feasibility and acceptability study, which will presumably be extended in a later, larger trial depending on the findings. The protocol clearly articulates the objectives of the study and its proposed conduct.

Authors' response

Thank you.

Page 6 line 24: what is known about the characteristics of online treatments for adolescent mental health problems that produce good adherence? How will those characteristics that promote high adherence be integrated into the proposed trial?

Authors' response

Evidence suggests that digital interventions incorporating some form of human support (e.g., via phone, texts, face-to-face visits) are more effective than purely self-administered platforms (Lehtimäki et al., 2021; Weisel et al., 2019). Often referred to as coaches or peer mentors, human supporters are not mental health specialists and do not deliver psychological interventions directly. Instead, they are trained lay workers who provide low-intensity and time-limited support to help users understand the digital intervention's content and overcome barriers to engagement.

Given this evidence, the Kuamsha app was designed to be supported by brief weekly phone calls from peer mentors. As explained on page 7: *“the primary role of the Peer Mentors is to support adherence and compliance with the app, troubleshoot problems related to the use of the app, and assist with the implementation of the homework activities. They will be trained not to provide additional advice or counselling and conduct their calls according to a pre-determined checklist of activities to help ensure fidelity”*.

In sub-Saharan Africa, task shifting using non-specialist workers has been increasingly employed to deliver physical and mental health treatment services (Galvin & Byansi, 2020; Singh & Sachs, 2013). However, limited research has investigated the potential for lay workers to supplement digital mental health interventions (Arjadi et al., 2018). The current study hopes to address this gap in the literature. As part of this work, a training platform was developed in collaboration with local experts and stakeholders in South Africa to guide the peer component of the intervention. This workstream is currently being tested, and the results from this piloting will be published as soon as it is available.

Section 2.2: this may go without saying, but I presume there is good mobile phone service coverage throughout the districts where the participants will be living?

Authors' response

There has been a proliferation of mobile phones in the area, but we are aware that at times mobile data coverage might not be fully reliable. To reduce potential problems with internet access during modules, both apps (treatment and control group) will be offered in both online and offline modes. The apps only require an internet connection when it is accessed for the first time, during which the app automatically downloads the modules to the device's internal storage. After that, the users can complete all modules at any other time, regardless of the internet signal.

In addition to this, and as explained on page 6, participants in both groups will receive 200MB of mobile internet data at different time points to ensure they have data to use the app, complete the online surveys, and can call/text for support.

One minor comment- suggest to include reference for PHQ score interpretation (page 7 line 50).

Authors' response

Thank you. We have included the reference for PHQ-A score interpretation, as follows (page 8):

PHQ-A scores of 5, 10, 15, and 20 represent mild, moderate, moderately severe and severe depression, respectively (Kroenke et al., 2001).

Reviewer: 3; Dr. Maria Bulgheroni, Ab.Acus srl, Milan, IT

The study is well designed and clear and repeatable (give the availability of similar digital tools).

Authors' response

Thank you.

I have a concern referring the relationship between the design and the objectives that is not fully clear to me. The primary objective of the study is "to determine feasibility and acceptability of a digitally delivered BA therapy intervention ...". I think that it is not clear enough what "digitally delivered BA intervention" means. Looking at the control and the intervention group, the main difference is not only the used app but also the support of trained Peer Mentors. The phone support plays a key role in the administration and follow up of the intervention. So, if this is the case (i.e. app+phone to be evaluated) it is important to clarify that the digital intervention consists not only of a digital application but also of a key phone support since the early phases of the manuscript. While, if the study aims at focusing on the digital app alone, it is needed to supply some form of phone support also to the control group. Please, clarify this issue in the manuscript.

Authors' response

Thank you, and apologies for the confusion. We have amended the manuscript to clarify that the digital intervention under evaluation includes both the Kuamsha app and the phone calls from peer mentors. This was described in *Section 2.5: Intervention arm Kuamsha programme* (page 6), as follows: "*The intervention arm will receive six modules of Behavioural Activation (BA) therapy via a smartphone application (the Kuamsha app) supported by trained Peer Mentors,*

implemented over 10 weeks. The app and phone calls from the Peer Mentors together comprise the Kuamsha programme”.

However, this clarification is also now part of the abstract as follows (page 2):

The treatment group will receive Behavioural Activation therapy via a smartphone application (the Kuamsha app) supported by trained peer mentors.

A second point refers to the method for assessing the digital competencies needed to run the digital app (inclusion criterion 3: is it only a matter of language knowledge of digital alphabetization? you are going to deliver them a new phone that they are not used to use and this may become a barrier; and exclusion criterion 6: what do you mean by capacity to use the app?). A not homogeneous digital capability of the participants may impact on the reliability of the results of the study. Please, clarify the way your are going to assess digital capabilities at the recruitment time.

Authors' response

Thank you for raising these important points.

Regarding digital competencies required to use the app: Central to this study was an iterative process to co-design the digital intervention through extensive formative work with adolescents from the study site. In total, more than 170 adolescents were involved in the development process. We also recruited other key stakeholders, including caregivers of adolescents, schoolteachers, and community leaders. During this formative work we were able to assess adolescents' familiarity with smartphones and ensure that the intervention was easy to use (usable) by all adolescents in the study context provided they are i) able to read in Xitsonga and ii) do not have a moderate to severe intellectual disability.

Inclusion criteria 3 states that adolescents should be able to “read sufficiently in the local language (Xitsonga) to use the Kuamsha app” (Page 5). This is because the app is a narrative (story-based) game and therefore includes Xitsonga text.

Exclusion criteria 6 states that participants will be excluded if they “lack capacity to consent to treatment or research participation or to use the app”. This criterion refers to adolescents who lack intellectual capacity (i.e. have a moderate to severe intellectual disability) and are therefore unable to consent to treatment or research participation or to use the app.

To assess these criteria we i) ask adolescents whether they can read in Xitsonga in our screening survey and ii) corroborate this information with the fieldworker who conducts the informed consent and baseline assessment with the adolescent as well as the adolescents' parent/ guardian.

Given that the app is easy to use by all adolescents who can i) read in Xitsonga and ii) do not have a moderate to severe intellectual disability, and that our formative work confirmed that adolescents in the study site are highly familiar with digital technology and smartphone devices, we did not think it necessary to assess the digital capabilities of adolescents at the time of recruitment. The results of the formative work will be published elsewhere.

References

- Arjadi, R., Nauta, M. H., Scholte, W. F., Hollon, S. D., Chowdhary, N., Suryani, A. O., Uiterwaal, C. S. P. M., & Bockting, C. L. H. (2018). Internet-based behavioural activation with lay counsellor support versus online minimal psychoeducation without support for treatment of depression: a randomised controlled trial in Indonesia. *The Lancet Psychiatry*, *5*(9), 707–716. [https://doi.org/10.1016/S2215-0366\(18\)30223-2](https://doi.org/10.1016/S2215-0366(18)30223-2)
- Avery, K. N. L., Williamson, P. R., Gamble, C., Francischetto, E. O. C., Metcalfe, C., Davidson, P., Williams, H., Blazeby, J. M., Blencowe, N., Bugge, C., Campbell, M., Collinson, M., Cooper, C., Darbyshire, J., Dimairo, M., Doré, C., Eldridge, S., Farrin, A., Foster, N., ... Treweek, S. (2017). Informing efficient randomised controlled trials: exploration of challenges in developing progression criteria for internal pilot studies. *BMJ Open*, *7*(2), e013537. <https://doi.org/10.1136/BMJOPEN-2016-013537>
- Bhana, A., Rathod, S. D., Selohilwe, O., Kathree, T., & Petersen, I. (2015). The validity of the Patient Health Questionnaire for screening depression in chronic care patients in primary health care in South Africa. *BMC Psychiatry*, *15*(1), 118. <https://doi.org/10.1186/s12888-015-0503-0>
- Cholera, R., Gaynes, B. N., Pence, B. W., Bassett, J., Qangule, N., Macphail, C., Bernhardt, S., Pettifor, A., & Miller, W. C. (2014). Validity of the patient health questionnaire-9 to screen for depression in a high-HIV burden primary healthcare clinic in Johannesburg, South Africa. *J. Affect. Disord.*, *167*, 160–166. <https://doi.org/10.1016/j.jad.2014.06.003>
- Diamond, A. (2013). Executive Functions. *Annu. Rev. Psychol.*, *64*(1), 135–168. <https://doi.org/10.1146/annurev-psych-113011-143750>
- Eilert, N., Wogan, R., Leen, A., & Richards, D. (2022). Internet-Delivered Interventions for Depression and Anxiety Symptoms in Children and Young People: Systematic Review and Meta-analysis. *JMIR Pediatr Parent* 2022;5(2)E33551 <https://Pediatrics.Jmir.Org/2022/2/E33551>, *5*(2), e33551. <https://doi.org/10.2196/33551>

- Galvin, M., & Byansi, W. (2020). A systematic review of task shifting for mental health in sub-Saharan Africa. *Int. J. Ment. Health*, *49*(4), 336–360. <https://doi.org/10.1080/00207411.2020.1798720>
- Huguet, A., Miller, A., Kisely, S., Rao, S., Saadat, N., & McGrath, P. J. (2018). A systematic review and meta-analysis on the efficacy of Internet-delivered behavioral activation. *J. Affect. Disord.*, *235*, 27–38. <https://doi.org/10.1016/J.JAD.2018.02.073>
- Kroenke, K., Spitzer, R. L., & Williams, J. B. W. (2001). The PHQ-9: Validity of a Brief Depression Severity Measure. *J. Gen. Intern. Med.*, *16*(9), 606. <https://doi.org/10.1046/J.1525-1497.2001.016009606.X>
- Lehtimäki, S., Martić, J., Wahl, B., Foster, K. T., & Schwalbe, N. (2021). Evidence on Digital Mental Health Interventions for Adolescents and Young People: Systematic Overview. *JMIR Ment Heal.*, *8*(4), e25847. <https://doi.org/10.2196/25847>
- Linardon, J., & Fuller-Tyszkiewicz, M. (2020). Attrition and adherence in smartphone-delivered interventions for mental health problems: A systematic and meta-analytic review. *J. Consult. Clin. Psychol.*, *88*(1), 1–13. <https://doi.org/10.1037/CCP0000459>
- Marlow, M., Skeen, S., Grieve, C. M., Hons, B., Carvajal, L., Åhs, J. W., Kohrt, B. A., Requejo, J., Stewart, J., Henry, J., Goldstone, D., Kara, T., & Tomlinson, M. (2022). Detecting Depression and Anxiety Among Adolescents in South Africa: Validity of the isiXhosa Patient Health Questionnaire-9 and Generalized Anxiety Disorder-7. *J. Adolesc. Heal.*, *0*(0). <https://doi.org/10.1016/J.JADOHEALTH.2022.09.013>
- Naslund, J. A., Aschbrenner, K. A., Araya, R., Marsch, L. A., Unützer, J., Patel, V., & Bartels, S. J. (2017). Digital technology for treating and preventing mental disorders in low-income and middle-income countries: a narrative review of the literature. *The Lancet Psychiatry*, *4*(6), 486–500. [https://doi.org/10.1016/S2215-0366\(17\)30096-2](https://doi.org/10.1016/S2215-0366(17)30096-2)
- Osborn, T. L., Venturo-Conerly, K. E., Wasil, A. R., Schleider, J. L., & Weisz, J. R. (2019). Depression and Anxiety Symptoms, Social Support, and Demographic Factors Among Kenyan High School Students. *J. Child Fam. Stud.*, 1–12. <https://doi.org/10.1007/s10826-019-01646-8>
- Patel, V., Weobong, B., Weiss, H. A., Anand, A., Bhat, B., Katti, B., Dimidjian, S., Araya, R., Hollon, S. D., King, M., Vijayakumar, L., Park, A. La, McDaid, D., Wilson, T., Velleman, R., Kirkwood, B. R., & Fairburn, C. G. (2017). The Healthy Activity Program (HAP), a lay counsellor-delivered brief psychological treatment for severe depression, in primary care in India: a randomised controlled trial. *Lancet*, *389*(10065), 176–185. [https://doi.org/10.1016/S0140-6736\(16\)31589-6](https://doi.org/10.1016/S0140-6736(16)31589-6)
- Ridley, M., Rao, G., Schilbach, F., & Patel, V. (2020). Poverty, depression, and anxiety: Causal evidence and mechanisms. *Science*, *370*(6522), eaay0214. <https://doi.org/10.1126/science.aay0214>
- Schueller, S. M., & Torous, J. (2020). Scaling evidence-based treatments through digital mental health. *Am. Psychol.*, *75*(8), 1093–1104. <https://doi.org/10.1037/AMP0000654>
- Singh, P., & Sachs, J. D. (2013). 1 million community health workers in sub-Saharan Africa by 2015. *Lancet*, *382*(9889), 363–365. [https://doi.org/10.1016/S0140-6736\(12\)62002-9](https://doi.org/10.1016/S0140-6736(12)62002-9)
- Snyder, H. R. (2013). Major depressive disorder is associated with broad impairments on neuropsychological measures of executive function: A meta-analysis and review. *Psychol. Bull.*, *139*(1), 81–132. <https://doi.org/10.1037/a0028727>
- Weisel, K. K., Fuhrmann, L. M., Berking, M., Baumeister, H., Cuijpers, P., & Ebert, D. D. (2019). Standalone smartphone apps for mental health—a systematic review and meta-analysis. *Npj*

VERSION 2 – REVIEW

REVIEWER	Mueller-Weinitschke, Claudia University of Freiburg, Department of Rehabilitation Psychology and Psychotherapy, Institute of Psychology
REVIEW RETURNED	01-Dec-2022
GENERAL COMMENTS	Thank you for the thorough review of the points previously noted. No further comments.
REVIEWER	Foulds, James A Department of Psychological Medicine, University of Otago
REVIEW RETURNED	16-Nov-2022
GENERAL COMMENTS	No further comments
REVIEWER	Bulgheroni, Maria Ab.Acus srl, Milan, IT, R&D
REVIEW RETURNED	01-Dec-2022
GENERAL COMMENTS	I would like to thank the authors for the modifications and explanations. I am fine with the current version of the manuscript. I still suggest to investigate as much as possible the link to digital abilities of the adolescents by using the collected information. It is always a key factor in acceptance.

VERSION 2 – AUTHOR RESPONSE

Reviewer 1; Ms Claudia Mueller-Weinitschke, University of Freiburg

Comments to the Author:

Thank you for the thorough review of the points previously noted. No further comments.

Authors' response

Thank you for your further review of our manuscript.

Reviewer: 2; Dr. James A Foulds, Department of Psychological Medicine, University of Otago

Comments to the Author:

No further comments

Authors' response

Thank you for your further review of our manuscript.

Reviewer: 3; Dr. Maria Bulgheroni, Ab.Acus srl, Milan, IT

Comments to the Author:

I would like to thank the authors for the modifications and explanations. I am fine with the current version of the manuscript. I still suggest to investigate as much as possible the link to digital abilities of the adolescents by using the collected information. It is always a key factor in acceptance.

Authors' response

Thank you for your further review of our manuscript.

We thank Reviewer 3 for the suggestion to consider adolescents' digital capabilities as a factor that could affect acceptance/ acceptability of the intervention. We agree this is important and therefore, as mentioned on page 7 of the manuscript, we will be assessing the acceptability of the intervention through the use of i) an acceptability questionnaire conducted at the end of the intervention assessment (week11) with all participants, as well as ii) in-depth interviews with a sub-sample of participants.

The acceptability questionnaire includes items which ask specifically about whether the app is easy to use for young people in this area. The interview guide also includes questions on whether adolescents found it easy/ challenging to use the app as well as which components/ parts of the app were easy/ challenging to use and why. The study team thinks that these questions will address the issue of digital capabilities of adolescents in this context, as such we have not edited our manuscript further.